# Supporting the Diagnosis of Fabry Disease Using a Natural Language Processing-Based Approach

**DOI:** 10.3390/jcm12103599

**Published:** 2023-05-22

**Authors:** Adrian A. Michalski, Karol Lis, Joanna Stankiewicz, Sylwester M. Kloska, Arkadiusz Sycz, Marek Dudziński, Katarzyna Muras-Szwedziak, Michał Nowicki, Stanisława Bazan-Socha, Michal J. Dabrowski, Grzegorz W. Basak

**Affiliations:** 1Saventic Health, Polna 66/12 Street, 87-100 Torun, Poland; ad.michalski2@gmail.com (A.A.M.); michal.kloska@saventic.com (S.M.K.); arkadiusz.sycz@saventic.com (A.S.);; 2Department of Analytical Chemistry, Nicolaus Copernicus University Ludwik Rydygier Collegium Medicum, 85-089 Bydgoszcz, Poland; 3Department of Hematology, Transplantation and Internal Medicine, Medical University of Warsaw, 02-097 Warsaw, Poland; 4Department of Pediatrics, Hematology and Oncology, Nicolaus Copernicus University Ludwik Rydygier Collegium Medicum, 85-094 Bydgoszcz, Poland; 5Department of Forensic Medicine, Nicolaus Copernicus University Ludwik Rydygier Collegium Medicum, 85-067 Bydgoszcz, Poland; 6Faculty of Mathematics and Information Science, Warsaw University of Technology, 00-662 Warsaw, Poland; 7Department of Hematology, Institute of Medical Sciences, College of Medical Sciences, University of Rzeszow, 35-959 Rzeszow, Poland; 8Saventic Foundation, Polna 66/12 Street, 87-100 Torun, Poland; 9Department of Nephrology, Hypertension and Kidney Transplantation, Medical University of Lodz, 90-419 Lodz, Poland; 10Department of Internal Medicine, Faculty of Medicine, Jagiellonian University Medical College, 31-008 Krakow, Poland; 11Computational Biology Group, Institute of Computer Science of the Polish Academy of Sciences, 01-248 Warsaw, Poland

**Keywords:** clinical diagnosis support system, decision-support, electronic health record, EHR, Fabry disease, natural language processing, NLP, rare disease, risk factor

## Abstract

In clinical practice, the consideration of non-specific symptoms of rare diseases in order to make a correct and timely diagnosis is often challenging. To support physicians, we developed a decision-support scoring system on the basis of retrospective research. Based on the literature and expert knowledge, we identified clinical features typical for Fabry disease (FD). Natural language processing (NLP) was used to evaluate patients’ electronic health records (EHRs) to obtain detailed information about FD-specific patient characteristics. The NLP-determined elements, laboratory test results, and ICD-10 codes were transformed and grouped into pre-defined FD-specific clinical features that were scored in the context of their significance in the FD signs. The sum of clinical feature scores constituted the FD risk score. Then, medical records of patients with the highest FD risk score were reviewed by physicians who decided whether to refer a patient for additional tests or not. One patient who obtained a high-FD risk score was referred for DBS assay and confirmed to have FD. The presented NLP-based, decision-support scoring system achieved AUC of 0.998, which demonstrates that the applied approach enables for accurate identification of FD-suspected patients, with a high discrimination power.

## 1. Introduction

Fabry disease (FD) is an ultra-rare genetic disorder with an incidence of 1:40,000 to 1:117,000, depending on the studied population [1]. It is caused by pathological variants of the alpha-galactosidase A gene (*GLA*) encoding lysosomal enzyme alpha-galactosidase A [2]. Lower activity of that enzyme leads to pathological accumulation of glycosphingolipids, such as globotriaosylceramides inside lysosomes, followed by cell degeneration, tissue inflammation, and organ impairment [3]. Since FD is an X-linked disorder, the symptoms are usually more severe in males than in females [4,5]. Interestingly, screening studies have revealed that the frequency of *GLA* pathological variants in newborns is much higher than the prevalence of clinically significant FD cases [6,7]. This phenomenon is likely a result of the complex gene expression profile or post-transcriptional modifications opposing the clinical role of the genetic *GLA* abnormalities [8,9,10]. Usually, the first clinical symptoms of FD appear early, during the preschool or school periods [11]. They include neuropathic pain (e.g., acroparesthesias), with the so-called “pain crises”, gastrointestinal symptoms, impaired sweating, and cold/heat intolerance. Other characteristic FD traits include recurrent fever, typical skin changes (e.g., angiokeratoma), ophthalmologic (e.g., cornea verticillata) and otolaryngological (e.g., hearing impairment or loss, dizziness) abnormalities, while those most frequently responsible for shortening the patient’s life include cardiac dysfunction, cardiovascular and cerebrovascular events, and chronic kidney disease, usually with proteinuria [12,13]. Despite many symptoms, diagnosis of FD in relation to the appearance of the first symptoms is usually significantly delayed (by 7–10 years), because most of the symptoms are non-specific [14,15,16]. Furthermore, awareness about this disease in the medical community is low [16,17]. It has been shown that 43% of pediatricians and 70% of rheumatologists could not point to the proper diagnostic approach in FD suspicion [16]. On the other hand, since effective therapy exists [18], early diagnosis is critical to limit disease progression; however, damage done to the organs prior to its initiation cannot be reversed [19,20,21]. Automatization of the screening process, based on approaches evaluating medical file records for suspicion of FD trait, might facilitate and shorten the time to diagnosis, benefiting patients and their physicians. Several applications have been described in the literature based on artificial intelligence (AI) or related methods for identifying rare diseases. For example, Hughes et al. [22] presented an approach helping to determine the severity of FD using an expert algorithm based on symptoms and clinical findings. Jefferies et al. [23] described an AI tool (OM1 Patient Finder™, OM1 Inc., Boston, MA, USA) capable of identifying patients at high FD risk. They proved the tool to be effective, with an AUC of 0.82. Access to large-scale databases of EHRs presents new opportunities to apply NLP methods and new sources of clinically valuable knowledge. One of the text processing tasks is the recognition of entities, especially in the medicine domain entities like symptoms or signs of certain conditions in general. Today’s implementations of NLP on unstructured narrative texts from EHRs have shown promising results on this task [24]. There is an increasing interest in BERT-like transformers [25,26]. Rule-based approaches have also been shown to be effective [24,27,28,29]. However, we decided to use a lexicon-based approach similar to that used by Oladapo Oyebode [29], but instead of the sentiment score, we used sentence embedding, implemented in spaCy [30], and cosine similarity to measure the level of similarity to achieve a complete automated process of extracting data from electronic health records (EHRs). It is important to note that medical description accounts for a substantial part of EHRs, and in many cases the majority of the information is in the form of unstructured data; therefore, NLP-based methods may be more sensitive than approaches based on ICD-9- or ICD-10-derived clinical phenotypes or on laboratory test results alone. Our study aimed to develop and test a risk-factor-based scoring system to support physicians in early diagnosis of FD through automatic screening and analysis of EHRs available in the information systems of primary care and outpatient clinics in real life with the use of NLP. Specific rare disease indication is much more efficient when a patient’s description is mapped against thousands of other records. In addition, tremendous amounts of complicated patient background information—i.e., the complete documentation of the patient’s medical history, which facilitates diagnosis—cannot be evaluated by a physician during one visit, in contrast to the presented approach. We believe that the NLP-based approach presented in this paper could accelerate accurate diagnosis among patients with signs of FD. 

## 2. Materials and Methods 

### 2.1. Studied Population

The study was conducted on medical data obtained from the Saventic Health database that covered eight Polish hospitals that had signed agreements with Saventic Health. It covers years 2008–2022, and all medical information of the patients were included in the study. However, in terms of the time horizon of the medical history, 25% of patients had less than 1 year of historical documentation, 50% less than 5 years, and 75% less than 9 years, respectively. The longest time range for a patient was 16 years between the first and last visit. This database was based on the PostgreSQL Relational Database and contained medical records that had been anonymized in accordance with the GDPR guidelines. The following types of data in unstructured form were present in the patients’ electronic health records (EHRs): age and gender, medical history, symptoms and signs, laboratory results, diagnoses, ICD-10 codes, imaging study descriptions, and epicrisis. Patient data were extracted and processed in Jupyter Notebook [31] in Python [32]. The Numpy [33] and Pandas [34] libraries were used for data preprocessing. The informed consent of the patients was not needed to perform analysis based on medical data from health care centers. Data were anonymized by the medical center prior to sharing data with the Saventic. After obtaining the results of the analysis (see Section 3), medical coordinators in individual facilities were contacted and the report specifying high-risk medical records was shared with them. The decision on whether to invite patients for further consultations, including a DBS, was taken by doctors leading individual patients. At first, it was verified, based on the presence of the ICD-10 code E75.2 in the EHRs, that 13 of the patients in the medical database had previously been diagnosed with FD using dry blood spot (DBS) assay. The EHRs of these 13 patients were obtained from the seven Polish hospitals at which they were treated. Together, these patients formed the study group. The control group was established with the aim of imitating a hospital population, in which this kind of scoring system could be implemented. For this reason, the control group was obtained from one hospital and included 19,372 patients from various hospital wards: allergology, angiology, surgery, phoniatrics, gastroenterology, hematology, cardiology, neonatology, neurology, neurosurgery, ophthalmology, otolaryngology, gynecology, obstetrics, rheumatology, urology, anesthesiology, internal medicine, emergency medicine, orthopedics and outpatients. None of the patients in the control group had been diagnosed with FD, and there was no information about FD in their EHRs. Unfortunately, the control group could potentially include undiagnosed FD patients. Patients were selected from the Saventic Health database according to inclusion and exclusion criteria listed below. (i) Inclusion criteria: age 18–75; no other criteria were defined in order to imitate implementation of the decision-support scoring system in a hospital population. (ii) Exclusion criteria: palliative care, alcohol dependency, disseminated cancer, blood cancer, and chemotherapy.

### 2.2. FD Risk Factor Development

The goal of the presented approach was to develop a scoring system for FD diagnosis based on digitized data from hospital records and its practical implementation. Firstly, according to detailed literature screening [23,35,36] and consultations with physicians from FD knowledge centers, a set of FD-related signs available in the literature was brought to 13 clinical features, considered by physicians as most useful and typical for FD. Secondly, based on expert knowledge, a scoring system for 13 clinical features was created. The score of each feature ranged from zero to three. When creating a scoring system, the following aspects were taken into consideration: the number of traits and characteristics included in a particular feature; the significance of a feature in FD diagnosis and/or severity; and the typical age of a feature’s first manifestation. Therefore, when a feature was common in the general population at a specific age, its score was low. On the other hand, if a feature typical for the elderly was observed in younger patients, e.g., stroke, myocardial infarction, renal failure, eye fundus lesions, or hearing loss, the scoring weight increased the score value of a feature. The sum of the clinical feature scores of a patient constituted the FD risk factor. Next, some of the clinical features typical for FD may be caused by other diseases known to be relevant for a patient. In such cases, when another confirmed disease was a potential source of a clinical feature, then it was not scored. In order to achieve this, several feature exclusion criteria were considered (Figure 1). One exclusion criterion was concerned with renal failure: nephropathies, amyloidosis, sarcoidosis, Alport’s syndrome, post-nephrectomy condition (single kidney, bilateral), metastatic urolithiasis, and congenital renal system defects. Another exclusion criterion was related with the clinical features of stroke: patent foramen ovale (PFO) and trauma. Finally, the 13 clinical features selected were further categorized according to the five most common FD signs: (1) cardiovascular symptoms; (2) kidney disease; (3) skin changes in selected areas; (4) neurological disorders; and (5) eye changes (Figure 1).

### 2.3. Natural Language Processing

NLP was implemented based on spaCy’s pl_core_news_md model [30]. The name *pl_core_news_md* consists of four elements: *pl* stands for the language on which the model was trained; *core* stands for general purpose, that is, tagging, parsing, lemmatization, and named entity recognition; *news* stands for the type of text data (blogs, news, comments) on which the model was originally trained by the spaCy developers; and finally, *md* stands for medium package size. This size has a reduced word vector table with 20 thousand unique vectors, resulting in approximately 500 thousand words. In our study, NLP was used to evaluate text records in the EHRs to obtain detailed patient characteristics (Figure 2). Regular expressions were combined to describe a medical term (Figure 2, step 1). Then, extensive medical descriptions found in the EHRs were broken down into sentences (Figure 2, step 2). The next step was to correct errors and typos occurring in the sentences, as well as to perform lemmatization (Figure 2, step 3). The text prepared in this way was subjected to tokenization (Figure 2, step 4). Tokenization is the division of sentences into meaningful units, usually separated in the text by white space. Regular expressions created in step 1 were searched for in the token sequence (Figure 2, step 5). Tokens were evaluated for the presence of a given medical term or its negation (Figure 2, step 6). This stage was supervised and usually required several hundred analyzed descriptions before it was considered to have worked properly, and the term saved in the dictionary (Figure 2, step 7). Subsequently, the cosine similarity of the specified token to an expression that already exists in the dictionary was checked (Figure 2, step 8). Finally, the medical term was extracted and evaluated, which contributed to the feature score (Figure 2, step 9). In cases where a token was related to a medical term that already existed and was stored in the dictionary, steps 6 and 7 were omitted. To determine how well the extraction technique performed, precision-oriented tests were conducted. Symptom-wise, 100 medical descriptions were analyzed, in which the presence of a symptom was detected; then, the correctness of the findings was manually verified by previously trained annotators. The results varied among symptoms, but they all exceeded the 70% precision threshold, at least. The model did not identify abbreviations if they were not in the dictionary.

### 2.4. FD Risk Score Implementation

To assess the FD risk factor among the studied population of patients, their EHRs were extracted from hospital systems. In a first step, ICD-10 codes were turned into their full descriptions, which allowed the extraction of additional patient information. Next, with the use of NLP, detailed patient characteristics were obtained from the text records. Then, the NLP-determined traits, laboratory test results, and characteristics obtained from the ICD-10 codes were assigned to 13 clinical features. Finally, using the established scoring system, each patient was assigned an FD risk factor. For the obtained values of risk factor for the patients from the study and control groups, a distribution plot was created and a cut-off value was established. The purpose of the cut-off value was to effectively reduce the number of patients referred for screening.

Several cut-off values were tested in order to choose the one that maximized the detection specificity of FD patients. None of the patients from the study group achieved a risk factor lower than three. For this reason, 3 was the lowest considered value for the cut-off. The risk factor values were verified between the cut-off of 3 and 11 in order to determine the specificity for the detection of FD patients. Based on the chosen cut-off value, patients were divided into those having low and high risk factor scores. The EHRs of those with high risk factor scores were further evaluated by two physicians, who decided whether a patient should be reported to the coordinator at the hospital at which the patient was being treated. There, the attending physician, having received the report, made a decision as to whether to contact the patient for further verification, e.g., to refer for DBS assessment, as is required for final FD confirmation [37,38]. At the final step of the analytical pipeline, a confusion matrix was created based on the established cut-off point. It was used to evaluate the number of patients with high or low risk factor scores among the total number of patients in the study and control groups. The quality of the FD risk factor scoring system was evaluated as follows: (1) accuracy, Equation (1); (2) precision, Equation (2); (3) recall, Equation (3); (4) F1-score, Equation (4); where *TN*—non-FD patients correctly classified as non-FD; *TP*—FD patients correctly classified as FD patients; *FP*—non-FD patients misclassified as FD patients; *FN*—FD patients misclassified as non-FD.
(1)$$\mathrm{Accuracy}=\frac{TP+TN}{TP+TN+FP+FN}$$
(2)$$\mathrm{Precision}=\frac{TP}{TP+FP}$$
(3)$$\mathrm{Recall}=\frac{TP}{TP+FN}$$
(4)$$F1\;\mathrm{score}=2\cdot \frac{\mathrm{Precision}\cdot \mathrm{Recall}}{\mathrm{Precision}+\mathrm{Recall}}$$

### 2.5. Statistical Analysis

Specific statistical tests were used to verify whether there was any evidence rejecting the null hypothesis, considering the attributes of the study and control groups. The null hypothesis was that there is no difference between the study and control groups. The Mann–Whitney test was used to compare continuous numerical variables, e.g., age, while the chi-squared test and Fisher’s exact test were used to compare categorical variables, e.g., sex proportions, incidence of cardiovascular diseases, skin changes, neurological disorders, kidney diseases, and eye disorders between the two analyzed groups. Statistics were calculated using the Sklearn [39] and Scipy [40] libraries, while the visualizations were created using Matplotlib [41] and Seaborn [42] libraries.

## 3. Results

This study presents a detailed NLP-based analysis of the EHRs of 19,385 patients. Their general characteristics and five most common FD signs are shown in Table 1. According to the obtained results, the prevalence of Fabry was higher among males than females (by 11.5%). The number of females and males was equal in the control group (within an accuracy of 1%). The statistical test failed to reveal any significant difference in terms of gender distribution between the study and control groups. Hence, there is not enough evidence to confirm a gender predisposition based on the observed differences among FD patients. In other words, patients are likely to have random chances rather than gender predisposition. On the other hand, the statistical test revealed a significant difference in mean age between the two groups, with FD patients being younger than patients in the control group. The results of the other tests showed that the distribution of traits differed significantly between both groups. As no eye disorders were observed in the study group, we excluded this sign from hypothesis testing. It was found that the most frequent sign in both the study and control groups was cardiovascular disease (Table 1) with the following clinical feature: hypertrophic.

The frequency of the 13 clinical features selected was scored to obtain the risk factor, and this varied between groups (Figure 3). Of all of the clinical features taken into account as part of the FD risk factor, the most frequent in the study group was myocardial infarction (Figure 3), while in the control group, it was hypertrophic cardiomyopathy (Figure 3).

The risk factor was individually computed for each patient from the study and control groups based on the NLP analysis performed on their EHRs (Figure 4). Using the obtained results, a distribution of the number of patients with a given risk factor was plotted. The patients with confirmed FD in the study group were aligned following the ranking, showing high risk score values (Figure 4).

### Patients with Diagnosed FD

Based on the distribution of the number of patients with specific risk factor values, several cut-offs were examined. For the assumed cut-off of 3, the specificity obtained was lower than when the cut-off was set to 4. At the same time, when assuming a cut-off equal to 5 or higher, the specificity increased, but the sensitivity dropped drastically. For this reason, a cut-off value equal to 4 was chosen. This effectively reduced the number of patients referred for screening. An adjustment was made for the risk factor of patients with confirmed FD, so that they would be included among those referred for screening. Moreover, the determined cut-off value accurately separated patients who should be referred for diagnostic tests and those whose symptoms were not alarming and/or had no signs of FD (Figure 4). In total, there were 92 patients who obtained a high risk factor ≥4. The risk factor distributions in the control and study groups are presented in Figure 5. 

Out of 92 patients with risk factors above the cut-off point, 80 were initially assigned to the control group, and their average risk factor was 4.3 (SD 0.62). The EHRs of these patients were verified by two physicians experienced in the diagnosis of FD, who came to the conclusion that of the 80 patients, 15 should be reported to the coordinators of the hospitals in which they were being treated, so that their attending physicians could decide whether to refer them for DBS assay for further diagnosis. At this point, it is known that one patient originally assigned to the control group, who received a high risk factor equal to 8 (Figure 4), was diagnosed with FD after a DBS assay was performed. Other patients received a negative test result. It is worth noting that there was one patient in the study group who was diagnosed with FD with a risk factor of 3, which is below the chosen cut-off point. The EHR of this patient was manually verified, and it was found that some symptoms, like skin changes, were described as: “Lesions of the type of multiple cavernous angiomas on the skin. Long-standing problem. Undiagnosed”, which, due to the general indications, was rated low in the scoring system, but if assigned more directly as “angiokeratoma” would have resulted in a higher risk factor assignment.

As mentioned, there were 92 patients with the risk factor ≥4. Among them, 80 patients were classified as FP and 12 were TP, which gave the ratio 1:7, while in the general population it is known to be 1:40,000 [1]. The obtained 1:7 ratio is 5714 times higher than the ratio represented within the general population. This represents significant knowledge gain. Next, a confusion matrix on all patients included in this study was prepared (Figure 6) to summarize the classification performance of the presented risk factor scoring for the cut-off set to 4. That evaluation helped to select patients who disagreed with the original class. The only FN patient known to have FD was confirmed to have a very general description in the EHR (as already mentioned). The returned confusion matrix (Figure 6) indicated an accuracy of 0.996, with sensitivity = 92.3% and specificity = 99.59%. However, due to the high class imbalance, precision = 0.1304, area under the precision–recall curve = 0.537 and F1-score = 0.2286 were found to be much more informative metrics. The obtained recall value supported the project’s assumptions. The low value of precision was anticipated, as from the beginning, it was assumed that some FD patients could be present in the control group. The high AUC value of 0.998 confirmed that the applied risk factor scoring and assigned cut-off point enabled accurate distinction between patient groups with a high discrimination power.

## 4. Discussion

Despite the low prevalence of rare diseases, their proper and early diagnosis, although challenging, is critical for affected patients. Therefore, a machine learning approach can accelerate their appropriate determination, e.g., in the case of FD [23,43,44], particularly when facilitated by NLP, as described in this paper. In this case, EHR analysis can highlight seemingly imperceptible facts or symptoms that typically elude physicians, preventing proper FD detection, especially in the case of physicians who have never seen the symptoms of this disease, e.g., cherry-like skin changes (angiokeratoma) coupled with pain in the extremities and recurrent fever. Furthermore, since FD symptoms develop over many years, long-term monitoring of medical history is essential [45]. Therefore, the long-term incorporation of AI algorithms may be a valuable tool for supporting physicians in their clinical practice. These algorithms will increase the chance of detecting typical symptoms, characteristic combinations thereof, and their changes over time, leading to correct diagnosis. Referring people with high risk factor, who might potentially suffer from FD, for diagnostic testing may be more effective and less expensive than screening the whole population. Moreover, AI algorithms that support physicians may also increase the awareness of rare diseases in the medical community. As a next step, they might also improve therapeutic approaches and prognostic decision-making processes in affected individuals. We presented a novel NLP-based approach that can support physicians in the diagnosis of FD based on the medical information included in patients’ EHRs, covering symptoms, laboratory results, and ICD-10 codes. The patients in the study group were 23–59 years old; thus, we decided that only adults would be included in the control group. In addition, the upper age limit in the control group was 75 years, in line with [46], where it was reported that more than 99% of 2044 FD patients had been diagnosed with FD before that age. As a result of this study, a narrow group of patients with a high risk factor of FD was identified. The likelihood of FD diagnosis in patients whose risk factor score exceeded the cut-off value was several thousand times higher than in the general population. The presented approach showed very high accuracy. However, accuracy might be a misleading metric in the case of imbalanced datasets where the searched trait is characterized by a very low incidence, as is the case with rare diseases. The disparity between the numbers of patients in the study and control groups was not related to data preparation, but to the fact that FD is a rare disease. For this reason, the AUC, precision, and F1-score served as performance metrics for the presented approach. In order to evaluate the effectiveness of the presented FD risk scoring system, a simple comparison was performed between the AUC values of the presented approach (AUC 0.998) and the approach designed by Jeffries et al. [23] (AUC 0.82). In the present study, the number of patients in the control group who had risk factor ≥4 was relatively small. Patients were assessed for a specific clinical presentation, and whether there were FN patients in the control group who did not undergo DBS assay remains uncertain. Therefore, it is difficult to judge what the true AUC was and whether it was really higher than the 0.82 presented by Jeffries et al. [23]. Their study [23] had significantly more participants, with close to 5000 FD patients in the study group and 1,000,000 patients in the control group, giving a ratio between study/control equal to 0.005. In our study, this ratio was 0.00067. Due to this fact, it is difficult to clearly determine which method is more effective. The presented approach is not a classic example of a classifier. Initially, the study group consisted of only 13 patients diagnosed with FD. Modeling such a small sample is difficult because of the high risk of overfitting due to the limited model verification capabilities. Test–train partitioning is affected by the sampling mechanism, and model learning and evaluation methods such as bootstrapping or leave-one-out cross-validation are computationally expensive. Models resulting from small-sample modeling are simple in structure, and factor-based models provide an alternative characterized by full controllability and explainability. Therefore, a risk-factor-based approach was used to select patients who should be referred for specialized diagnostic testing. The presented approach was developed to screen patients at the population scale. This would facilitate and accelerate the work of physicians. Moreover, the use of the presented approach could potentially reduce diagnostic costs, as it will indicate the need for diagnostic testing only for patients whose risk factor score exceeds the cut-off value.

The implementation of the presented approach contributed to the diagnosis of one patient with FD, as confirmed with DBS assay. This patient was 45, with a history of unexplained cardiomyopathy, vertigo, asymptomatic ischemic changes in the brain MRI, and renal transplant in a first-degree relative under 45, resulting in a risk factor of 8. At the time of the implementation of the risk factor-based approach, this patient was clinically suspected of amyloidosis, and FD was not considered as a potential reason at any point in the patient’s EHRs. Thus, our method led to the proper diagnosis, which is critical for the affected patient. Family screening also confirmed a diagnosis of FD in the patient’s brother. The approach presented in this paper makes it possible to narrow the pool of patients with a high risk factor who should be referred for DBS assay. Even imperfect indications can be perceived as beneficial due to the raising of physicians’ suspicions, followed by patients’ referring for further evaluation and proper diagnostic tests, therefore resulting in earlier diagnosis. However, it should be kept in mind that this method of risk factor assessment still has a chance of missing individual patients whose EHRs analysis incorrectly underestimates their risk factor. However, in such cases, further data follow-up may be decisive. The use of the approach presented in this paper may prove particularly useful due to the fact that it is able to draw attention to seemingly irrelevant facts and can combine multiple features (e.g., ICD-10 codes) to improve predictive and diagnostic capabilities. In recent years, we have observed a breakthrough in NLP competition, large pretrained transformers like BERT [47] and GPTs [48] have become state-of-art and general-purpose language models due to their capacity to contextualize word representations. This means that models like BERT can understand the meaning of words based on their context in a sentence, allowing it to capture more nuanced and complex relationships between words. It has been demonstrated that BERT-based models can be used effectively in medical tasks [49]. Unsupervised training generates embeddings used as inputs for downstream supervised tasks such as disease classification. They can significantly improve performance, particularly on smaller datasets where labeled data are limited. The vast majority of implements have considered only the English language; however, our research is based on content in Polish. To overcome this problem, procedures for transferring knowledge from multilingual to monolingual BERT-based models were developed, thanks to which HerBERT (Polish BERT) was developed [50]. Transformers constitute the state of the art, and can be considered to be a direction worth incorporating into our study to enhance or replace the NLP search approach we developed.

## 5. Conclusions

This work aimed to create a risk factor scoring system to support physicians in FD diagnosis using real-life data. The authors used an original approach to the NLP tool evaluating patients’ EHRs and assessing risk factor among patients who may suffer from FD. The proposed method increases the effectiveness of diagnostics, further improving patients’ quality of life and prognosis. To the best of the authors’ knowledge, this is the first case of FD diagnosis with the help of NLP application. Despite the promising results of this experiment, there are several limitations that need to be considered. First of all, the sample size of the FD patients was relatively small, and EHRs are incomplete, which may limit the generalizability of the conclusions drawn. Therefore, further development and testing of the risk factor with a larger and more diverse patient population are required to precisely assess usability in a clinical setting. Secondly, the control group could potentially include undiagnosed FD patients. However, for this reason, we present our solution as a screening tool at the population level, and not as a classifier in itself. Thirdly, the NLP algorithm implemented in this study requires further development to improve its accuracy. While the algorithm has demonstrated its utility, there is still room for improvement in terms of its ability to generalize and analyze context. Since the confirmation of the studies are prospective tests, an appropriate amount of time is needed to collect patients for the DBS examinations. These limitations highlight the need for continued research and development.

## Figures and Tables

**Figure 1 jcm-12-03599-f001:**
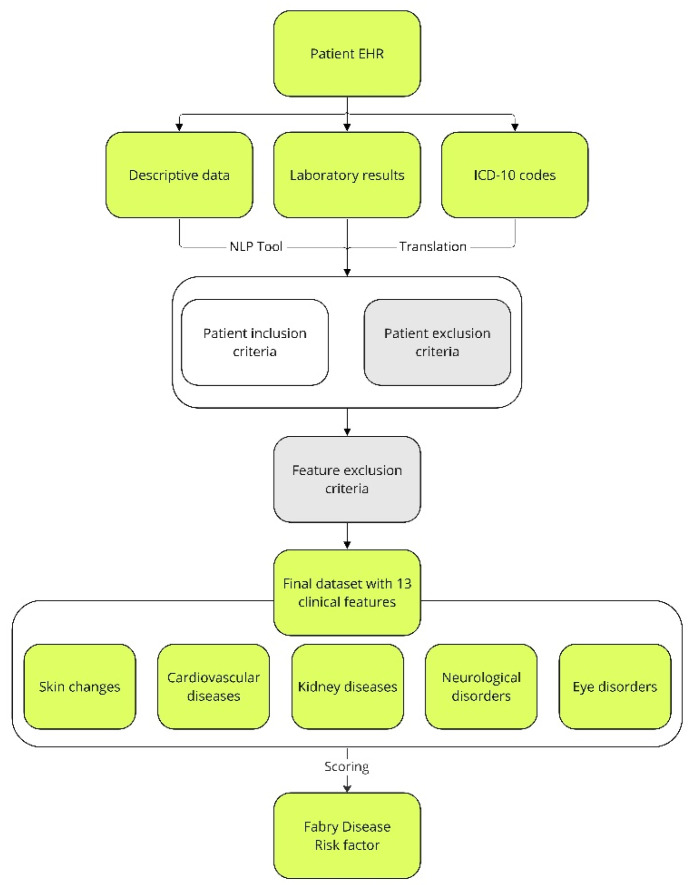
Overview of patients’ assignment to the study and control groups, as well as the extraction of characteristics related to FD from text records and descriptions of ICD-10 codes. The fields that were used for FD risk factor assessment are indicated in lime. Fields eliminating patients from further analysis due to specific symptoms or exclusion criteria, which are described in the Materials and Methods, are indicated in gray. EHR—electronic health records; ICD-10—International Statistical Classification of Diseases and Related Health Problems; NLP—natural language processing.

**Figure 2 jcm-12-03599-f002:**
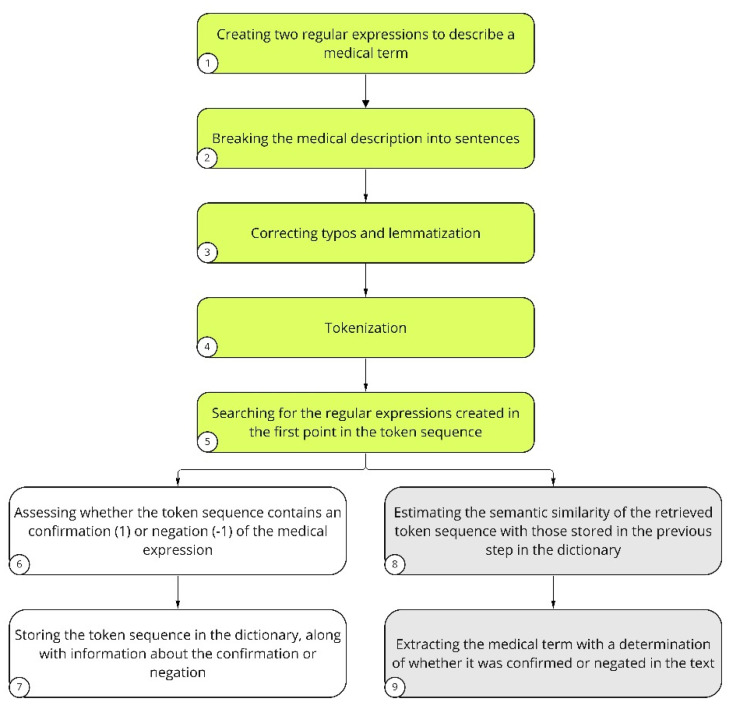
Description of NLP analysis performed on text records of the study and control patient groups. The steps of LP involving preparation of the patients’ EHRs to be assessed are indicated in green. The key NLP steps for introducing a new term to the dictionary are indicated in white. Medical terms that are already present in the dictionary are indicated in gray.

**Figure 3 jcm-12-03599-f003:**
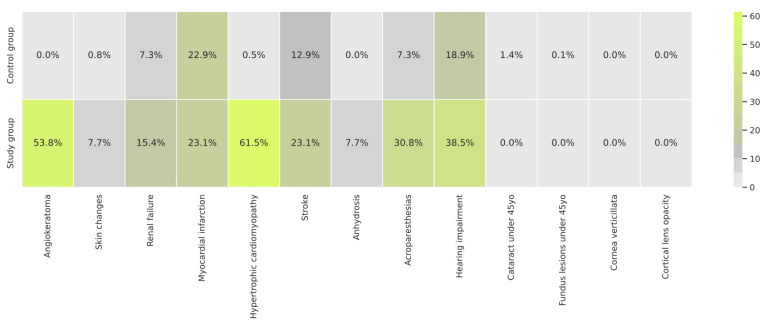
Frequency of the clinical features extracted from EHRs with the use of NLP in the study and control groups.

**Figure 4 jcm-12-03599-f004:**
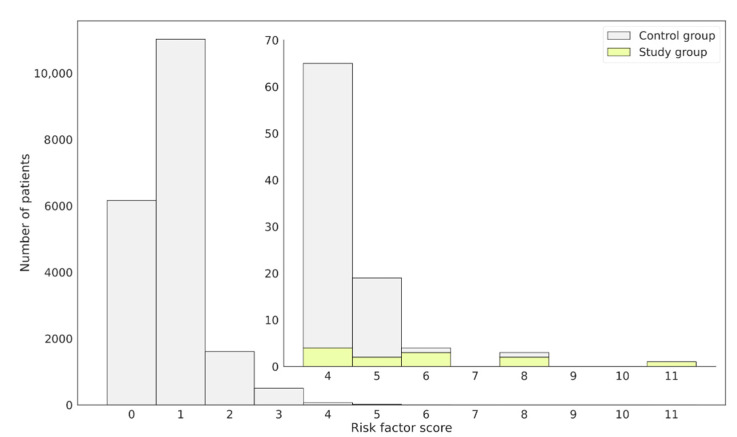
Distribution of patients assigned to the control or study group regarding the obtained FD risk factor score. Confirmed FD patients, constituting the study group, obtained high-risk score values and were placed on the right-hand side of the plot. The external bars represent the whole patient population, while the internal focus on those having the risk factor ≥4. Bars in yellow represent.

**Figure 5 jcm-12-03599-f005:**
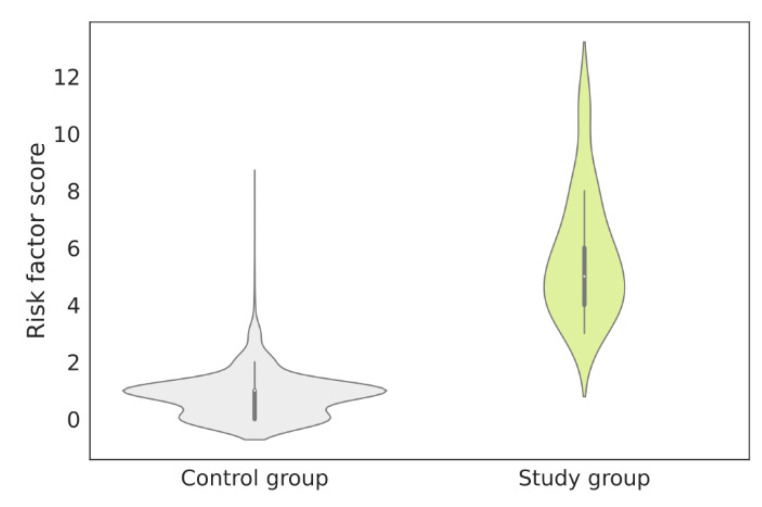
Risk factor distributions in the control group and the study group. The mean risk factors in the study group and the control group were 5.7 (SD 2.13) and 0.8 (SD 0.72), respectively. SD—standard deviation.

**Figure 6 jcm-12-03599-f006:**
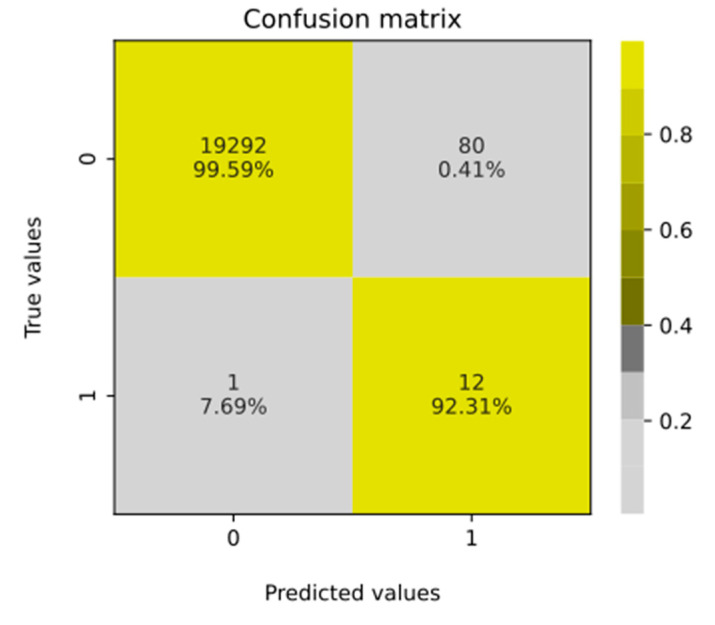
Confusion matrix computed for all patients including their FD risk factor values and cut-off value equal to 4. When interpreting the given values, it is important to remember that the issue concerns the prevalence of a rare disease (see Discussion).

**Table 1 jcm-12-03599-t001:** General characteristics and five most common FD signs, extracted with the use of NLP, presented for the patients from the study and control groups; Study group—patients with confirmed FD; Control group—patients presumably without FD. SD—standard deviation.

	Study Group (*n* = 13)	Control Group (*n* = 19,372)	*p*-Value
Sex (% female)	38.5%	50.5%	0.38
Mean age (SD)	45.2 (10.5)	55.5 (13.3)	*p* < 0.05
Cardiovascular diseases (%)	10 (76.9%)	6641 (34.3%)	*p* < 0.05
Skin changes (%)	8 (61.5%)	149 (0.8%)	*p* < 0.05
Neurological disorders (%)	7 (53.8%)	5034 (26.0%)	*p* < 0.05
Kidney diseases (%)	4 (30.8%)	2030 (10.5%)	*p* < 0.05
Eye disorder (%)	0 (0%)	298 (1.5%)	-

## Data Availability

The data presented in this study are available on request from the corresponding author. The data are not publicly available due to personal data and privacy protection.

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
