# Peer review of "Supporting the Diagnosis of Fabry Disease Using a Natural Language Processing-Based Approach"

_jcm, 2023, doi:10.3390/jcm12103599_

Round 1

Reviewer 1 Report

I would like to thank the authors for this contributions. The study presents an interesting application of NLP in the medical domain. The methodology is mostly clear, and the manuscript is well-written as well. However, please consider the comments below in the next version.

(1)

It's important to be cautious when interpreting the reported ROC-AUC due to the highly imbalanced nature of the dataset, as acknowledged by the authors. When the distribution of classes is not uniform, the ROC-AUC can be misleading, as a classifier can achieve a high score by simply predicting the majority class while performing poorly on the minority class. This can give a false impression of the model's performance. In such cases, it would be more appropriate to report the Precision-Recall curve instead. If the ROC-AUC is reported, the abstract should clearly state this issue, and the precision and recall values should be reported as well to provide a more complete picture of the model's performance.

(2)

The introduction should be properly positioned within the context of NLP for healthcare. It would be helpful to reference more recent studies that have discussed or implemented related applications in healthcare to provide a better understanding of the current state of the field.  For example:

https://doi.org/10.3390/healthcare10112270

https://doi.org/10.5220/0010414508250832

(3)

The references of spaCy and Scikit-Learn should be cited, please.

(4)

With the increasing adoption of transformer models (e.g. BERT) in healthcare or medical studies, I think this could be considered as a future direction for the present study. The authors may touch on that point as part of the future work.

Author Response

Reviewer1

Comment (1):

It's important to be cautious when interpreting the reported ROC-AUC due to the highly imbalanced nature of the dataset, as acknowledged by the authors. When the distribution of classes is not uniform, the ROC-AUC can be misleading, as a classifier can achieve a high score by simply predicting the majority class while performing poorly on the minority class. This can give a false impression of the model's performance. In such cases, it would be more appropriate to report the Precision-Recall curve instead. If the ROC-AUC is reported, the abstract should clearly state this issue, and the precision and recall values should be reported as well to provide a more complete picture of the model's performance.

Response:

The reviewer pointed out a significant element about imbalanced classes. Following the suggestion the  AUPRC will be provided in the updated version of the manuscript. However, as far as we follow this topic the AUROC is insensitive to changes in class distribution [reference: An introduction to ROC analysis by Tom Fawcett]. Unlike for example accuracy where as a class distribution changes measure will change as well, even if the fundamental classifier performance does not.

In the abstract there is a clear statement about obtained results: “The presented NLP-based, decision-support scoring system achieved  12 AUC of 0.9986, which demonstrates that the applied approach enables for accurate identification of  13 FD-suspected patients, with a high discrimination power”

Comment (2):

The introduction should be properly positioned within the context of NLP for healthcare. It would be helpful to reference more recent studies that have discussed or implemented related applications in healthcare to provide a better understanding of the current state of the field.  For example:

https://doi.org/10.3390/healthcare10112270

https://doi.org/10.5220/0010414508250832

Response:

We have followed the suggestion of the Reviewer and inserted into the Introduction section a paragraph about NLP in the healthcare context:

The access to large-scale databases of EHRs introduces new opportunities to apply NLP methods and new sources of clinically valuable knowledge. One of the text processing tasks is the recognition of entities, especially in the medicine domain entities like symptoms or signs of certain conditions in general. Today’s implementations of NLP on unstructured narrative texts from EHRs have shown promising results on this task [24]. We can observe increasing interest in BERT-like transformers [ 25 , 26]. Rule-based approaches have also been shown to be effective [24,27 –29 ].

Comment (3):

The references of spaCy and Scikit-Learn should be cited, please.

Response:

The references were added, please find in the newer version of the manuscript:

NLP was based on SpaCy’s pl_core_news_md model [30].

Comment (4):

With the increasing adoption of transformer models (e.g. BERT) in healthcare or medical studies, I think this could be considered as a future direction for the present study. The authors may touch on that point as part of the future work.

Response:

Following the reviewer suggestion, a new paragraph was added into the discussion section.

In a recent years we observed a breakthrough in NLP competition, large-pretrained

transformers like BERT [47] or GPTs [48] have come to be state-of-art and general purpose

language models due to their capabilities to contextualize word representations. This

means that models like BERT can understand the meaning of words based on their context

in a sentence, allowing it to capture more nuanced and complex relationships between

words. It has been demonstrated that BERT-based models can be effectively used in medical

tasks [49]. Unsupervised training generates embeddings used as inputs for downstream

supervised tasks such as disease classification. They can significantly improve performance,

particularly on smaller datasets where labeled data is limited. The vast majority of imple-

mentations concerned only the English language however our research is based on content

in Polish. To overcome this problem were developed procedures of transferring knowledge

from multilingual to monolingual BERT-based models thanks to which HerBERT (Polish

BERT) was developed [50]. Transformers constitute state-of-the-art and can be considered

as a direction worth incorporating into our study to enhance or replace the NLP search

approach we developed.

Reviewer 2 Report

REVIEW: Supporting the diagnosis of Fabry disease using a natural language processing-based approach

Summary: A clinical decision support tool was developed to improve diagnosis of Fabry disease, a rare disease with many non-specific symptoms. Therapy exists motivating early diagnosis so irreversible damage to organs occurs.

Abstract:

·       Another sentence or two about the risk score would be helpful – what were the predefined features or how did the features get decided (review and consensus from two or three clinicians, most common symptoms reported in the literature)?

·       Results on the score would be interesting as well, what was the score range and mean/median? Did you have cutoffs?

·       The sentence on line 9 “Then, medical records of patients with the highest FD risk score were revised by physicians who decided whether to refer a patient for additional tests or not” is confusing, did you mean reviewed and not revised?

o   Then also I read this as a retrospective study, so all patients would have had Fabry disease, was referral for more testing a blinded review by providers and part of the risk score testing or was this performed on new patients with FD like symptoms? A description of the study design and population would be helpful in the abstract.

Introduction:

·       Is there any evidence or speculation that automation through NLP would perform better than the other methods described in paragraph 3? What are drawbacks from the AI method or expert algorithm? Do these methods not allow for prospective diagnosis?

·       NLP may not be well known to many clinicians just yet, a short description of what it is and how it is used in clinical research would be helpful.

·       Lines 59-60 – it would be helpful to be more specific here – what background information are you discussing, just the patient’s record or records of many other patients?

·       The introduction is well written with a clear problem described, description of the disease and motivation for finding a solution to the problem.

Methods:

·       The study design should be stated early on in the methods

·       Line 69: “the following information was included in the electronic health record” this information is always present in an EHR. Did you mean to say that information was queried and extracted from the EHR?

·       What years were records extracted and how far back in each medical record did you search?

·       Line 82 should belong in the discussion section under study limitations

·       If the goal is to generate a risk score to be used in primary care and outpatient clinics, describe why was an entire hospital population used? I assume a hospital population includes many wards such as oncology or obstetrics that are unrelated to Fabry disease. List the wards considered for this study.

·       Why did you choose just one hospital and not match based on the hospital where each of the 13 patients came from?

·       Development of the scoring system (more detail is needed here): Literature screening was only 3 citations, was a systematic review ever conducted? How many physicians were consulted? How did you settle on 13 features, which I am reading as different from signs and symptoms (how were the features developed)? Was consideration given to symptoms that present earlier in the disease progression versus later? What are those 13 features? What features were considered but excluded and reasons for exclusion?

·       Paragraph lines 106-113 – needs some proofreading –

o   Line 106 - diagnosed should be diagnoses

o   Line 109 – why just “one set” suggest deleting or elaborating what how many sets of exclusion exist

o   Line 111-112 “these exclusion criteria were all concerned with renal failure

·       NLP – was there any measure of how well the extraction technique performed especially if it was trained for blogs, news and comments and not medical records? How did the model account for terminology abbreviations?

·       Statistical testing:

o   What was the null hypothesis (or multiple hypotheses) you tested?

o   Since the cases and controls were not matched did you control for anything?

o   The abstract mentions AUC, there is no mention of this in the risk score development or statistical testing.

o   Outside of excluding certain comorbidities, did you control for any others or at least look at common ones over the study population?

o   Describe the other comparisons made in the results – what did you do?

Results:

·       Table 1/Paragraph 1 should include more description/demographics about the study population

·       Paragraph 2, still unaware of the final 13 clinical FD features you considered so it is hard to interpret Figure 3

·       Line 191 – how did you choose those cutoffs?

·       Line 192-197 – share the sensitivity and specificity values

·       Line 197 – 199 – describe the symptoms that would trigger a referral for diagnostic testing, symptoms considered not alarming, and no signs of FD (score of 0).

·       Line 207-210 – it would make sense to publish results after you have the DBS findings from the other 13 patients so you can report those numbers.

·       Line 213-216 – I read this more as the NLP tool needs further training, you can’t expect every clinician to properly document

·       Line 218-221 – I would delete this, these comparisons are not the same and does not make sense for this analysis

·       Figure 6 and  Line – 226-234 – it would make more sense to complete this after you get the DBS back for those 13 controls with high risk scores, you don’t really know the false positive rate until you receive those results

General comment on results

·       Without any indication of duration of symptoms its hard to distinguish how useful this risk score is… did the symptoms present early that therapy could be effective?

·       Did you look at Kappa scores for the providers deciding whether to refer for testing?

Discussion

·       Line 266-269 – this is true! Did you consider looking at positive/negative predictive value which is sensitive to the prevalence of a disease? I would suggest presenting these stats alongside Sensitivity and specificity

·       Line 290-292 – this sounds like the main goal of the study, this should be stated in the introduction – key being referral for test and not actual diagnosis, also the statistic description should reflect this, it’s confusing to present accuracy results (one thinks of diagnosis and not necessarily referral), be specific when you are presenting your results

·       The authors only compared this study with one other, a second was mentioned in the intro, are there any others you found on risk scores for FD or even comparing to clinical guidelines?

·       Limitations section missing – a big limitation is the small number of patients with FD

Author Response

Reviewer 2

Comments and Suggestions for Authors

REVIEW: Supporting the diagnosis of Fabry disease using a natural language processing-based approach

Summary: A clinical decision support tool was developed to improve diagnosis of Fabry disease, a rare disease with many non-specific symptoms. Therapy exists motivating early diagnosis so irreversible damage to organs occurs.

Abstract:

Comment 1:·     

 Another sentence or two about the risk score would be helpful – what were the predefined features or how did the features get decided (review and consensus from two or three clinicians, most common symptoms reported in the literature)?

Response 1:

In the abstract only the general idea is presented “The NLP-determined elements, laboratory  6 test results, and ICD-10 codes were transformed and grouped into pre-defined FD-specific clinical  7 features that were scored in the context of their significance in the FD signs. The sum of clinical  8 feature scores constitutes the FD risk score.” We prefer to keep it as it is because of the length limitations required for the abstract. Further details about the risk factor itself are presented in the Methods section: “2.2. FD risk factor development”. It is shown there that the selection of the features was expert-knowledge based. The specific selected 13 features are presented in Figure 3, and the groups of features in the methods section point 2.2. and presented in Figure 1.

Comment 2:·     

 Results on the score would be interesting as well, what was the score range and mean/median? Did you have cutoffs?

Response 2:

The range of scores  in the study sample was 0 - 11. The most frequent score was 1 (approximately 57% of samples). Check Response 20 for details about cut-offs.

Comment 3:·      

The sentence on line 9 “Then, medical records of patients with the highest FD risk score were revised by physicians who decided whether to refer a patient for additional tests or not” is confusing, did you mean reviewed and not revised?

Response 3:

Thanks for the comment. That was confusing indeed. It was changed to “reviewed”.

Comment 4:·

Then also I read this as a retrospective study, so all patients would have had Fabry disease, was referral for more testing a blinded review by providers and part of the risk score testing or was this performed on new patients with FD like symptoms? A description of the study design and population would be helpful in the abstract.

Response 4:

The nature of the study is retrospective - in the current version we put such information in the Abstract. We find it impossible to put all suggested in the comment information into the Abstract and to hold the required 200 words. Because of that the study design is described in the Methods section and in the abstract we decided to add just a general statement about “retrospective nature of the study”: - “To support physicians, we developed a decision support scoring system created on the basis of a retrospective research.”

Introduction:

Comment 5:·      

Is there any evidence or speculation that automation through NLP would perform better than the other methods described in paragraph 3? What are drawbacks from the AI method or expert algorithm? Do these methods not allow for prospective diagnosis?

Response 5:

The aim to put multiple methods in the introduction was to introduce a reader into a broader context of the possible analysis - types and tools. Although, there are other publications that in detail compare the advantages and disadvantages of them. In our opinion that all is beyond the scope of this work. As the reviewer's comment points to a very interesting issue we decided to modify the introduction where we can speculate that NLP might perform better than other approaches due to the nature of the task. Much of the information in EHR is in the unstructured text written by doctors.  Therefore we can expect that the method based on an NLP engine might be more sensitive and specific than other approaches. Simply speaking, first the NLP has to be implemented to perform data mining.

We modified text:

“It is important to note that medical description consists substantial part of EHR, in many cases majority of information is in the form of unstructured data, therefore,  NLP base method may be more sensitive than the approaches based ICD - 9 , ICD - 10, ICD -9 derived clinical phenotypes or on laboratory test results alone. “

At the same time a new section in the results was added where we refer to data extraction performance:

“To know how well the extraction technique performed, the precision-oriented tests were conducted. Symptom-wise 100 medical descriptions were analyzed, in which the presence of a symptom was detected, then the correctness of the findings was manually verified by previously trained annotators. The results varied across symptoms but they all exceeded at least the 70% precision threshold. The model did not identify abbreviations if they were not in the dictionary.”

Comment 6: ·      

NLP may not be well known to many clinicians just yet, a short description of what it is and how it is used in clinical research would be helpful.

Response 6:

The topic about NLP was extended, now it is as follows:

Access to large-scale databases of EHRs introduces new opportunities to apply NLP methods and new sources of clinically valuable knowledge. One of the text processing tasks is the recognition of entities, especially in the medicine domain entities like symptoms or signs of certain conditions in general. Today’s implementations of NLP on unstructured narrative texts from EHRs have shown promising results on this task [24]. We can observe increasing interest in BERT-like transformers [25,26]. Rule-based approaches have also been shown to be effective [24,27–29]. However, we decided to use a lexicon-based approach similar to that used by Oladapo Oyebode [29] but instead of the sentiment score, we used sentence embedding implemented by spaCy [30] and cosine similarity to measure the level of similarity to achieve a complete automated process of extracting data from electronic health records (EHRs). It is important to note that medical description consists substantial part of EHR, in many cases majority of information is in the form of unstructured data, therefore, NLP base method may be more sensitive than the approaches based ICD-9, ICD-10 derived clinical phenotypes or on laboratory test results alone.

Comment 7:

Lines 59-60 – it would be helpful to be more specific here – what background information are you discussing, just the patient’s record or records of many other patients?

Response 7:

The sentence has been changed:

“In addition, a tremendous amount of complicated patient background information - complete documentation of medical history, which facilitates a diagnosis can not be evaluated by a physician during one visit, in contrast to the presented approach.”

The introduction is well written with a clear problem described, description of the disease and motivation for finding a solution to the problem.

Methods:

Comment 8:

The study design should be stated early on in the methods

Response 8:

In the current version we keep the classic layout consistent with the section title: Material and Methods. So, first we present the material that was taken for analysis, and then the methods with a simultaneous description of the stages of work - stages in accordance with the order of their execution. We understand that everyone may have their own perception of content and sometimes a different layout may be easier for someone to understand, however, since we create the risk factor first and then test it, we prefer to keep the layout as it currently is.

Comment 9:

 Line 69: “the following information was included in the electronic health record” this information is always present in an EHR. Did you mean to say that information was queried and extracted from the EHR?

Response 9:

The sentence could be misleading indeed. We changed the wording of this sentence. We wanted to show the variety of data types that were in EHRs and to point out their lack of structure.

“The following types of data in unstructured form were present in the patients’ electronic health records (EHRs):”

Comment 10:

 What years were records extracted and how far back in each medical record did you search?

Response 10:

The records are taken from the period between 2008-2022. We have taken full EHRs of patients without any time limitations. Whole descriptions.

The sentence clarifying that issue was added into the methods section:

The study was conducted on medical data obtained from the Saventic Health database that covered eight Polish hospitals that signed agreements with Saventic Health. It covers years 2008-2022 and all medical information of the patients were included in the study. However, in terms of the time horizon of medical history, 25% of patients have less than 1 year of historical documentation, 50% less than 5 years, and 75% less than 9 years, respectively.  The longest time range for a patient was 16 years between the first and last visit.

Comment 11:

 Line 82 should belong in the discussion section under study limitations

Response 11.

A new section: Conclusions was added into the manuscript and several limitations are shown there.

Comment 12:

 If the goal is to generate a risk score to be used in primary care and outpatient clinics, describe why was an entire hospital population used? I assume a hospital population includes many wards such as oncology or obstetrics that are unrelated to Fabry disease. List the wards considered for this study.

Response 12:

From the medical point of view, FD patients have problems directly connected to the pathophysiology of Fabry disease, however,  similarly to the general population, they  experience a whole spectrum of medical problems. Using NLP we are able to extract information from medical interviews, physical examinations diagnosed independently from the unit on which the patient is hospitalized. At least, in theory medical interview, physical examination should also include general information about the history of the patient i.e. information about past diagnosis, family medical history. 

We are aware of the fact that the number of diagnosed FD patients is underestimated. Symptoms and signs of FD may occur in EHRs of patients from multiple wards. We expect our risk factor to be implemented at the population scale that is why we used multiple wards to catch the EHRs heterogeneity. Information about wards was added into the Materials and Method section:

“Patients were obtained from the wards including : ”

-             Allergology

-             Angiology

-             Surgery

-             Phoniatrics

-             Gastroenterology

-             Hematology

-             Cardiology

-             Neonatology

-             Neurology

-             Neurosurgery

-             Ophthalmology

-             Otolaryngology

-             Gynecology, Obstetrics

-             Rheumatology

-             Urology

-             Anesthesiology

-             Internal Medicine

-             Emergency Medicine

-             Orthopedics

-             outpatients

 Detailed information about wards was added into the manuscript text.

Comment 13:

Why did you choose just one hospital and not match based on the hospital where each of the 13 patients came from?

Response 13:

We have chosen one hospital as it was a pilot project and we had good collaboration possibilities in this facility. Due to administrative challenges we were not able to start our research  at once in many medical centers.

We are not sure whether we have understood the question correctly, but following this how we got it: we believe that this is irrelevant from which hospitals positive FD patients were obtained. From the epidemiological standpoint it seems that the fact of undetermined FD across hospitals is probably similar. It is believed that in Poland there are around between 100 - 200  patients that remain undiagnosed. Because the hospital selected to deliver the control group has multiple wards, we believe that its heterogeneity is representative to other medical facilities. Information about facilities was added into the manuscript - please take a look at the response 12.

Comment 14:

Development of the scoring system (more detail is needed here): Literature screening was only 3 citations, was a systematic review ever conducted? How many physicians were consulted? How did you settle on 13 features, which I am reading as different from signs and symptoms (how were the features developed)? Was consideration given to symptoms that present earlier in the disease progression versus later? What are those 13 features? What features were considered but excluded and reasons for exclusion?

Response 14:

We conducted a literature review focusing on clinical presentation of Fabry disease. In creation of clinical phenotypes we focus on symptoms listed in the publication of Fabry Outcomes Survey. In the Fabry Outcomes Survey following categories of symptoms are listed: cardiological, neurological, ophthalmological , gastrointestinal, dermatological, vascular. Selection of the symptom was done in collaboration  with two physicians who have experience in taking care of the patient with Fabry disease, however, still it was a judgment call on our side. After analyses we decided to create

  1. angiokeratoma;
  2. Skich changes
  3. Renal failure/ proteinuria
  4. Myocardial infarction < 45
  5. Hypertrofic cardimyopaty
  6. Stoke
  7. Anhidrosis
  8. Acroparestsis
  9. hearing impairment
  10. Cataracta under 45
  11. Fundus lesion under 45
  12. Cornea verti…
  13. Cortical lens opacity

We understand clinical features as symptoms that can be defined using NLP, ICD - 10, ICD - 9  or exclusion features.

Symptoms were used to refer to “terms” related with a disease - most general in our approach. Clinical feature is the most detailed element - entity for which the NLP model was searching. Sign is the name of a term that contributes to multiple clinical features. The interrelationships and hierarchy between the words might be taken from the context -

see abstract:

The NLP-determined elements, laboratory  6 test results, and ICD-10 codes were transformed and grouped into pre-defined FD-specific clinical  7 features that were scored in the context of their significance in the FD signs.

see methods:

Firstly, according to detailed literature screening [23,35,36] and consultations with physicians from FD knowledge centers, a set of FD related signs available in the literature was brought to

13 clinical features, considered by physicians as most useful and typical for FD.

Finally, the selected 13 clinical features were further grouped into five most common FD signs: 1) cardiovascular symptoms, 2) kidney disease, 3) skin changes in selected areas, 4) neurological disorders, and 5) eye changes (Fig. 1).

Finally, in our work, we did not analyze the course of the disease as such. We checked whether the patient had features that, taken together, indicate that he/she could have FD. But the issue suggested in the comment seems interesting for us and we will try to address it in our further studies.

Comment 15:

Paragraph lines 106-113 – needs some proofreading –

o   Line 106 - diagnosed should be diagnoses

o   Line 109 – why just “one set” suggest deleting or elaborating what how many sets of exclusion exist

o   Line 111-112 “these exclusion criteria were all concerned with renal failure

Response 15:

The pointed paragraph was rewritten:

“Next, some of the clinical features typical for FD may be caused by other diseases known for a patient. In such cases, when another confirmed disease was a source of a clinical feature, it was not scored. To achieve that, several feature exclusion criteria were considered (Fig. 1). Exclusion criteria concerned with renal failure: phropathies, amyloidosis, sarcoidosis, Alport’s syndrome, post-nephrectomy condition (single kidney, bilateral), metastatic urolithiasis, and congenital renal system defects. Another exclusion criteria were related with a stroke clinical feature: patent foramen ovale (PFO) and trauma.”

Comment 16:

NLP – was there any measure of how well the extraction technique performed especially if it was trained for blogs, news and comments and not medical records? How did the model account for terminology abbreviations?

Response 16.

Thanks for the interesting comment. We find it relevant and because of that a new fragment into the Methods section was added. Just to note - the SpaCy’s pl_core_news_md model was trained on the blogs, news, comments. That was done by the developers of that model. In our methods section we provide explanation of the model name. In our approach the model training was done using medical descriptions. The paragraph about SpaCy was changed so as not to cause such misunderstandings.

New text that was added to the current version of the ms: “To know how well the extraction technique performed, the precision-oriented tests were conducted. Symptom-wise 100 medical descriptions were analyzed, in which the presence of a symptom was detected, then the correctness of the findings was manually verified by previously trained annotators. The results varied across symptoms but they all exceeded at least the 70% precision threshold. The model did not identify abbreviations if they were not in the dictionary.”

Comment 17:

  • Statistical testing:

o   What was the null hypothesis (or multiple hypotheses) you tested?

Response 17a: The null hypothesis was that there is no difference between study and control group.

o   Since the cases and controls were not matched did you control for anything?

Response 17b: Univariate testing has shown significant statistical differences between both groups for most feature distributions. The fundamental assumption of experiment is that patients' EHRs in both groups are different in terms of describing attributes. Differences are expected and desirable in the sense of modeling FD. Conducted calculations confirmed this assumption for most of the patient traits. We did not perform correction for any batch effects.

o   The abstract mentions AUC, there is no mention of this in the risk score development or statistical testing.

Response 17c: Risk score was arbitrarily fixed, AUC was calculated post-factum. Statistical testing is oriented towards group dissimilarities in terms of features with fixed class labels.

o   Outside of excluding certain comorbidities, did you control for any others or at least look at common ones over the study population?

Response 17d:

The seven physicians who participated in the research (see list of authors) have carefully gone through the EHRs of the whole control group and they did not find any pattern that should be reported.

o   Describe the other comparisons made in the results – what did you do?

Response 17e:

Shortly speaking, the performed comparisons are named in the section of statistical analysis. In the results section they are presented in Table 1. In the current version of the manuscript the comparisons are meticulously presented.

“According to the obtained results, the prevalence of Fabry was higher among males than females (by 11.5%). The number of females and males is equal in the control group (within accuracy of 1%).  The statistical test failed to reveal any significant difference in terms of gender distribution between study and control groups. Hence, there is not enough evidence to confirm a gender predisposition based on the observed differences among FD patients. In other words, patients are likely to have random chances rather than a gender predisposition. On the other hand, the statistical test revealed a significant difference in mean age between both groups, FD patients are younger than the control group. The results of other tests have shown that the distribution of traits differs significantly between both groups. As no eye disorders were observed in the study group, we excluded this sign from hypothesis testing.            It was found that the most frequent sign in both the study and control groups was  “Cardiovascular diseases” comprising the following clinical features: hypertrophic cardiomyopathy, stroke, and myocardial infarction. Out of all clinical features taken into  account in the FD risk factor, the most frequent in the study group was myocardial in- farction (Fig. 3) while in the control group it was hypertrophic cardiomyopathy (Fig. 3).“

Results:

Comment 18:

Table 1/Paragraph 1 should include more description/demographics about the study population

Response 18:

Of course, the more accurate the demographic description of the groups, the better, but it should be remembered that when using descriptions (EHRs) of very different patients, we had to deal with a situation in which many features were present in a very small number of patients. In such a situation of very low sample size of patients with a specific feature the reasoning power would be very small. That is why we decided that medical doctors will decide - select the features to be presented in the table and we would like to keep it that way. In the current version of the manuscript the time range from which the descriptions come was added.

At the same time, it seems to us that the features relevant to the subject have been presented. If the comment would be more precise we could add the pointed features.

Comment 19:

 Paragraph 2, still unaware of the final 13 clinical FD features you considered so it is hard to interpret Figure 3

Response 19:

The paragraph was changed to directly say that all selected features n=13 that are included in the risc score are presented in Fig. 3.

The added sentence into paragraph two is: The frequency of selected 13 clinical features scored to obtain the risk factor varied between groups (Fig. 3).

Comment 20:

Line 191 – how did you choose those cutoffs?

Response 20.

The cut-off points were searched manually by trial and error in such a manner to achieve the best ratio between specificity and sensitivity.

In the methods section it was explained: For the obtained values of risk factor for the patients from the study and control 147 groups, the distribution plot was created and a cut-off value was established. The purpose 148 of the cut-off value was to effectively reduce the number of patients referred for screening. 149 Several cut-off values were tested to choose the one that maximizes detection specificity of 150 FD patients. None of the patients from the study group obtained a risk factor lower than 151 three. That is why three was the lowest considered value for the cut-off. The cut-off from 3 to 11 risk factor 153 values verified 152 the specificity for FD patients detection. Based on the defined cut-off value, patients were divided into those having low-risk 154 and high-risk factor score. EHRs of those with a high-risk factor were further evaluated  155 by two physicians who decided whether a patient should be referred for DBS assessment, 156 required for final FD confirmation ([37,38]).

In the results section it was written: Based on the distribution of the number of patients with specific risk factor values, several cut-offs were examined. For the assumed cut-off equal to three the obtained specificity was lower than for four. At the same time, for the assumed cut-off equal to five or higher the specificity increased but the sensitivity dropped drastically.

Comment 21:

Line 192-197 – share the sensitivity and specificity values

Response 21.

Sensitivity=92.3 and specificity=99.59%, you can read it directly from the confusion matrix (Fig. 6). This information was added into the results section.

Comment 22:

Line 197 – 199 – describe the symptoms that would trigger a referral for diagnostic testing, symptoms considered not alarming, and no signs of FD (score of 0).

Response 22:

FD has very non-specific symptoms, as described in the introduction. Therefore, there is no unambiguous simple division into features that clearly indicate the disease or exclude it. It would be more appropriate to talk here about a set of rules where each of the features in the rule has its own weight, and rules where the significance of individual features may change the weight provided that another feature is present in the rule. It is an extremely complex system, we are trying to develop it using machine learning tools, but due to the small number of patients, we do not have enough information to effectively teach such an AI model. This is a very broad topic, we do not want to discuss it here because this work would become multithreaded. What we're dealing with here is the risk factor, which gives us a ranking of patients. It is a filter that should help physicians perform screening analyses.

Comment 23:

 Line 207-210 – it would make sense to publish results after you have the DBS findings from the other 13 patients so you can report those numbers.

Response 23:

We do have all the results right now. Out of 15 patients who were decided to take a DBS one obtained positive result and additionally later it was confirmed that his brother has also FD. The text was changed.

Comment 24:

 Line 213-216 – I read this more as the NLP tool needs further training, you can’t expect every clinician to properly document

Response 24.

That's right, the tool still needs to be developed to extend the lexicon and minimize error originating from improper or unusual documentation. We mentioned it in the “Conclusions” section.

Comment 25:

Line 218-221 – I would delete this, these comparisons are not the same and does not make sense for this analysis

Response 25.

Here we can not agree with the comment. The general ratio in the population shows what is the proportion of having a FD, and we show how more likely it is - that we gain knowledge if our scoring system will score a patient. That is a great increase.  Let’s say that the proposed model is a diagnostic test. This line refers to the test precision, since among 92 patients selected by this model 12 of them were TP (odds that given a positive test are 1:7). Assuming that the experiment reflects the general hospital population, we can expect precision to be proportional between both populations. On the other hand, without our test the odds of having FD are assumed to be 1:40 000. Therefore, conducting the test gives us an informational gain. However, estimation of FD prevalence is itself vague, authors of cited articles reported prevalence in the range of 1:40 000 to  1:117 000.

Comment 26:

Figure 6 and  Line – 226-234 – it would make more sense to complete this after you get the DBS back for those 13 controls with high risk scores, you don’t really know the false positive rate until you receive those results

Response 26:

We agree. Currently we have it and the text was changed.

General comment on results

Comment 27:

Without any indication of duration of symptoms its hard to distinguish how useful this risk score is… did the symptoms present early that therapy could be effective?

Response 27:

We agree that duration of a symptom is important as well as to map in the patient history when a symptom appeared first. We are aware of it, but it was not implemented at the beginning of our study. Currently we are building NLP models which will be able to select such information from unstructured EHRs. Still in our opinion the risk score might be useful as its main aim is to perform filtering at the population scale and selection of these patients who because of the high risk score should be examined in detail by physicians.

This issue will be implemented into the next algorithm.

Comment 28:

 Did you look at Kappa scores for the providers deciding whether to refer for testing?

Response 28.

Well, first of all, physicians only reviewed patients selected by the algorithm so we can't measure agreement in rejections. If we consider only lets call it “partial agreement”: the accepted/selected fraction which is 15/80 we will get a relatively low agreement. That is why in our opinion Kappa score would not be practical in this case. We could use it to measure the agreement among physicians decisions, but they worked as a consilium. We will apply it in our further studies.

Discussion

Comment 29:

Line 266-269 – this is true! Did you consider looking at positive/negative predictive value which is sensitive to the prevalence of a disease? I would suggest presenting these stats alongside Sensitivity and specificity

Response 29.

You can find answer to this question in the “Results” section:

The returned confusion matrix (Fig. 6) indicated accuracy at the level of 0.996. However, due to high class imbalance, precision = 0.1304, recall = 0.9231, area under precision-recall curve=0,537 and F1-score = 0.2286 were found to be much more informative metrics.

Comment 30:

 Line 290-292 – this sounds like the main goal of the study, this should be stated in the introduction – key being referral for test and not actual diagnosis, also the statistic description should reflect this, it’s confusing to present accuracy results (one thinks of diagnosis and not necessarily referral), be specific when you are presenting your results

Response 30:

Starting from the manuscript title where it is written “Supporting the diagnosis” through multiple paragraphs in the text, each time we underline that this is a risk factor based on the scoring system which may support the diagnosis - filter patients who should be considered for further tests, who should be verified by physicians whether in their case a FD might be considered.

There are multiple examples where we put it in the text:  in the abstract it is written in one place:

To support physicians, we developed a decision support scoring system

And in the second place:

The sum of clinical feature scores constitutes the FD risk score. Then, medical records of patients with the highest FD risk score were reviewed by physicians who decided whether to refer a patient for additional tests or not.

And at the end of the abstract:

The presented NLP-based decision-support scoring system achieved  AUC of 0.998, which demonstrates that the applied approach enables for accurate identification of FD-suspected patients, with a high discrimination power.

In the introduction

Automatization of the screening process, based on approaches evaluating medical file records for suspicion of FD trait, might facilitate and shorten the time to diagnosis, benefiting patients and their physicians.

and in the other place of the introduction:

Our study aimed to develop and test a risk factor-based scoring system that supports physicians in the FD early diagnosis through automatic screening and analysis of EHRs available in the information systems of primary care and outpatient clinics in real life with the use of NLP.

In our opinion the main aim of the research is well pronounced.

Comment 31:

 The authors only compared this study with one other, a second was mentioned in the intro, are there any others you found on risk scores for FD or even comparing to clinical guidelines?

Response 31:

To the author's best knowledge at the moment of manuscript preparation these were the only published papers.

Comment 32:

Limitations section missing – a big limitation is the small number of patients with FD

Response 32.

You can find described limitations in the “Conclusions” section.

Despite the promising results of the experiment, there are several limitations that need to be considered. First of all, the sample size of the FD patients was relatively small and EHRs are incomplete which may limit the generalizability of the drawn conclusions. Therefore, further development and testing of the risk score with a larger and more diverse patient population are required to precisely assess usability in a clinical setting. Secondly, the NLP algorithm implemented in the study requires further development to improve its accuracy. While the algorithm has proven its utility, there is still room for improvement in terms of its ability to generalize and analyze context. Since the confirmation of the studies are prospective tests, the appropriate time is needed to collect patients for the DBS examinations. These limitations highlight the need for continued research and development.

Reviewer 3 Report

Michalski Adrian A, et al.

Supporting the diagnosis of Fabry disease using a natural language processing-based approach

Dr. Michalski AA and colleagues submitted a paper with a title:” Supporting the diagnosis of Fabry disease using a natural language processing-based approach”. Authors present a very interesting new approach using an artificial intelligence (AI) method in order to identify patients at risk for having Fabry disease (FD). Data from the literature clearly showing that in daily clinical practice the diagnosis of FD is still delayed and numbers of FD patients in different countries are still far from estimated prevalence.

A new method of NLP approach seems very logical, although personally I’m not able and don’t have enough knowledge to comment the technical part and solutions, but first results are very promising.

Although I’m very strongly supporting the paper, there are still some important issues to be solved or addressed.

1.       Are affiliations number 2 and 7 the same? They are very similar. Under number 2 is Saventic Health and under number 7 is Saventic Foundation.

2.       Introduction:

a.       Line 36 and 37 – please clarify the sentence: “…diagnosis of FD is usually significantly delayed (usually 7-10 years)…..”. Namely it is not clear according to what is diagnosis delayed. Probably to the first symptom? (line 205-209)

3.       Results:

a.       There is a discrepancy in results and outcomes. In the Abstract authors stated that one patient who obtained a high-FD risk score was referred for DBS assay and confirmed to have FD. While in the Results they stated that two patients were diagnosed with FD.

b.       According to the text (line 205-209) at the end 15 patients were referred to DBS assay for further diagnosis. At the moment it is known that two are already confirmed for FD, but others are still in the process of making a DBS assay appointment.  It is not logic to me to publish a paper and results, without actually finishing the study. These, final results are very important and every patient (esp. the positive one) counts.

c.       This is related to additional comments by authors interpreting the results that out of 90 patients with high risk score, there were 12 Fabry patients. But, as mentioned previously two patients from control group were already proven to be positive for Fabry disease. And again analysis is not finished yet. With already diagnosed FD patients and with potentially additional patients from additional analysis the total number of patients would increase and the final result for the method used would be even much better.

d.       I do not agree with the decision to include 15 patients selected by physicians for DBS assay in the TP group (line 237). Namely diagnosis of FD must be confirmed according to international standards and not on documentation valuation. These 15 patient should be properly diagnosed and then further analysis can be made.

4.       General:

a.       It is not clear if and how the patients gave an informed consent

b.       Since the females with FD are much less affected, it would be interesting to see analysis of the NLP method specifically for each gender.

c.       Phenotype of the FD male patients much depends on mutation. Please include information on mutations (classic or late-onset) and also submit a list of mutations (possible as Suppl.). Namely one would expect that patients with classic mutations would have a much higher risk score. Of course all those analyses should be interpreted with care due to the small numbers and this fact should be included in limitations.

d.       I would suggest to finish diagnostic analysis in all patients who were referred to further diagnosis in order to get the final number of FD patients and then to re-submit the paper.

Author Response

Reviewer 3

Comment: 1.      

Are affiliations number 2 and 7 the same? They are very similar. Under number 2 is Saventic Health and under number 7 is Saventic Foundation.

Response:

Thank you for pointing out this issue. There is no mistake; Saventic Health is a company that mostly concentrates on new algorithms development and whole population screening whereas Saventic Foundation focuses on helping people who suffer from some disease that, despite many years of diagnosis, has not been identified. These people may upload their documentation and foundation supports the diagnosis process, among others, by accelerating contact with specialists in the field. That is why the co-authored physicians are affiliated to the Saventic Foundation and data scientist Saventic Health.

Comment: 2.      

Introduction:

  1. Line 36 and 37 – please clarify the sentence: “…diagnosis of FD is usually significantly delayed (usually 7-10 years)…..”. Namely it is not clear according to what is diagnosis delayed. Probably to the first symptom? (line 205-209)

Response:

Following the suggestion the sentence was changed: “Despite many symptoms, diagnosis of FD in relation to the appearance of the first symptoms  is usually significantly delayed by 7-10 years, because most of them are non-specific ([14–16]).”

Comment: 3a.      

Results: There is a discrepancy in results and outcomes. In the Abstract authors stated that one patient who obtained a high-FD risk score was referred for DBS assay and confirmed to have FD. While in the Results they stated that two patients were diagnosed with FD.

Response:

Thanks for pointing this out. This mistake appeared because of taking into account information from subsequent reports from hospitals and updating subsequent versions of the manuscript. There was one confirmed patient.

Comment: 3b.      

According to the text (line 205-209) at the end 15 patients were referred to DBS assay for further diagnosis. At the moment it is known that two are already confirmed for FD, but others are still in the process of making a DBS assay appointment.  It is not logic to me to publish a paper and results, without actually finishing the study. These final results are very important and every patient (esp. the positive one) counts.

Response:

 In the case of our study, we are not dealing with a situation where a certain number of patients enter the study. The context of our research work is population-based. As part of cooperation with medical units, we have access to electronic descriptions of patients' medical history. But these data are pseudo-anonymized and we have no contact with the patient. After performing the screening analyses, expert physicians verify the results of the algorithm and if they confirm the suspicion of the disease, then such information is transferred to the given public health service unit. It is the coordinating physician who contacts the patient to make an appointment for DBS, but the patient may refuse, may have changed his place of residence and cannot be contacted. We are aware that this is not an ideal study plan, but due to the nature of cooperation with medical centers, it is the only one to be implemented in our opinion. We cannot determine when all patients will be verified in time, when we provide such information to medical units.

At this moment all reports from the health clinics were returned and currently we report the results of all 15 patients. One was confirmed positive and it turned out that also his brother is FD positive. The text of ms was changed.

Comment:

This is related to additional comments by authors interpreting the results that out of 90 patients with high risk score, there were 12 Fabry patients. But, as mentioned previously two patients from control group were already proven to be positive for Fabry disease. And again analysis is not finished yet. With already diagnosed FD patients and with potentially additional patients from additional analysis the total number of patients would increase and the final result for the method used would be even much better.

Response:

The reviewer referred to an important element. In the previous response to the comment, we described in more detail the form of organization of our research. Due to the passage of time devoted to the processing of our manuscript, we have already collected a complete set of feedback from all medical facilities and these results have been presented in the paper.             

Comment:

 I do not agree with the decision to include 15 patients selected by physicians for DBS assay in the TP group (line 237). Namely diagnosis of FD must be confirmed according to international standards and not on documentation valuation. These 15 patient should be properly diagnosed and then further analysis can be made.

Response:

You are right, this paragraph is misleading and does not provide any relevant insight therefore it was removed in the updated version of manuscript.

Comment: 4.     

 General: a.       It is not clear if and how the patients gave an informed consent

Response:

Due to the nature of the legal agreements we have with medical units, we have access to the medical record  of patients who have been in a given medical unit in the last 14 months. Granting such access is in accordance with Polish law and on our part the data is pseudo-anonymised and protected on many levels. As a result, bioinformatics analyzes on data can be performed without the patient's knowledge. However, as a result, after obtaining the analysis result, we contact the hospital and only someone from the hospital has the right to contact the patient and invite them for further consultations. With such a constitution of cooperation, the consent of patients is not needed. The downside of this solution is that the patient may refuse to come for further consultations or the leading physician may not agree with the algorithm assessment.

In the Materials and Methods section it was added:

The informed consent of the patients was not needed to perform analyzes based on medical data from health care centers. Data was anonymized by the medical center prior to sharing data with the Saventic. After obtaining the results of the analysis, medical coordinators in individual facilities were contacted and the report specifying a high risk medical records was shared with them. Decision on whether to invite patients for further consultations, including a DBS, was taken by doctors leading individual patients.

Comment: b.      

Since the females with FD are much less affected, it would be interesting to see analysis of the NLP method specifically for each gender.

Response:

We have followed the suggestion and in the current version of the manuscript in the first paragraph there are presented results associated with this issue. The first paragraph is as follows:

“This study covered a detailed NLP-based analysis of EHRs of 19,385 patients. Their general characteristics and five most common FD signs were shown in Table 1. According to the obtained results, the prevalence of Fabry was higher among males than females (by 11.5%). The number of females and males is equal in the control group (within accuracy of 1%).  The statistical test failed to reveal any significant difference in terms of gender distribution between study and control groups. Hence, there is not enough evidence to confirm a gender predisposition based on the observed differences among FD patients. In other words, patients are likely to have random chances rather than a gender predisposition. On the other hand, the statistical test revealed a significant difference in mean age between both groups, FD patients are younger than the control group. The results of other tests have shown that the distribution of traits differs significantly between both groups. As no eye disorders were observed in the study group, we excluded this sign from hypothesis testing. ”

Comment: c.      

Phenotype of the FD male patients much depends on mutation. Please include information on mutations (classic or late-onset) and also submit a list of mutations (possible as Suppl.). Namely one would expect that patients with classic mutations would have a much higher risk score. Of course all those analyses should be interpreted with care due to the small numbers and this fact should be included in limitations.

Response:

The issue raised by the reviewer is very important although we do not have access to such specific information and it was not covered by our analysis. Data of patients with Fabry disease do not include information about mutation. Since this information is incorporated in that data after diagnosis of the patients in our studies it is not crucial.  Data of the patient positive group was gathered from the whole medical record in the Polish clinics. In most clinics genetic data is not digitized.  Also due to anonymisation procedures in the places where all genetic data is digitized we do not get this data form the medical centers.

Comment: d.      

I would suggest to finish diagnostic analysis in all patients who were referred to further diagnosis in order to get the final number of FD patients and then to re-submit the paper.

Response:

Due to the passage of time, we have already collected complete feedback from hospitals and it is presented in the results section.

Round 2

Reviewer 1 Report

Thanks, I have no further comments.

Reviewer 3 Report

Dear authors,

I have to congratulate you on an excellent paper. You gave all the answers and corrections according to my review and I strongly believe your paper will have an important impact on the further development of similar projects.